# Uncovering Resilience Disparities among Religious Groups in Israel’s Mass COVID-19 Vaccination Drive: Lessons Learned to the Post-COVID Era

**DOI:** 10.3390/bs13050398

**Published:** 2023-05-10

**Authors:** Miri Sarid, Maya Kalman-Halevi, Rony Tutian, Sharon Gilat-Yihyie, Adi Sarid

**Affiliations:** 1Department of Education, Western Galilee College, Acre 2412101, Israel; mayah@wgalil.ac.il (M.K.-H.); ronyt@wgalil.ac.il (R.T.); sharong@wgalil.ac.il (S.G.-Y.); 2Sarid Institute, Haifa 2626047, Israel; adi@sarid-ins.co.il

**Keywords:** anxiety, COVID-19, Israel, religious groups, resilience, social support

## Abstract

The aim of the current study was to examine the emotional resilience, satisfaction with life, social support, and anxiety during the vaccination process of the Israeli population after the end of the third lockdown, according to religiosity degree. We hypothesized that a higher degree of religiosity (ultra-Orthodox and religious participants) would be associated with higher levels of resilience and with lower levels of anxiety than in secular individuals. In addition, it was hypothesized that satisfaction with life, social support, anxiety, and religiosity will predict resilience and anxiety. Nine hundred and ninety-three native Jewish Hebrew-speaking respondents representing ultra-Orthodox, religious, observant, and secular Jews participated in this study. Ultra-Orthodox participants showed higher resilience and satisfaction with life than other groups, and lower levels of anxiety. Satisfaction with life and social support predicted higher resilience. It is suggested that religious faith as well as satisfaction with life may provide a source of strength and resilience in stressful life events.

## 1. Introduction

In early 2021, Israel initiated a groundbreaking mass vaccination campaign aimed at reducing the spread of COVID-19. By March of the same year, approximately 53.2% of the population had received their first vaccine dose, and 40.9% had received both doses [1]. With the decline in infection rates and the end of the third lockdown, the Israeli population began to resume their regular daily activities. At the point in time that the current study was conducted, the Israeli population was relieved from the health threat due to the vaccine and an increase in the economic crisis posed previously by the complete closure forced on the population [2]. At this time, we were interested in exploring emotional resources (e.g., satisfaction with life (SWL), social support), religion, and socioeconomic status (SES), and their relationship with resilience and anxiety in understanding the process of recovery and returning to routine among four groups of varying degrees of religiosity in Israel.

Resilience is based on theoretical foundations from various disciplines such as mental health, psychology, medicine, and sociology [3]. Researchers use the terms “adaptation or development”, “positive adjustment”, or “recovery or growth from adverse conditions” [4,5]. It is a normative response to crisis and trauma [4,6,7] and a long-term outcome of the current crisis of the COVID-19 pandemic. Highly resilient individuals can successfully overcome trauma and crisis and return to their routines [8]. On the other hand, low resilience may lead to mental health issues such as anxiety, depression, and substance use [9]. Resilience plays a crucial role in mitigating negative mental effects during traumatic and uncertain situations, thereby improving an individual’s well-being [10]. Religiosity can serve as a significant resource in promoting mental health, providing a framework for meaning-making and a source of psychological well-being [11]. In Israel, the Jewish population comprises various cultural groups, distinguished by degrees of religiosity. Among them are the ultra-Orthodox, a unique cultural group that prioritize religious law (”halacha”) over governmental law. However, during the COVID-19 outbreaks in Israel, the ultra-Orthodox were disproportionately affected, with a significantly higher infection rate compared to the secular population [12]. Despite accounting for only 10% of the population in Israel [13], they made up approximately one-third of confirmed cases during the peak of the second wave [14].

The ultra-Orthodox community is known for their strong social capital and shared belief in the protective power of physical gatherings such as prayer services, religious school classes, and community rituals [15]. This social capital can explain their high life expectancy and good mental and physical health [16]. While this way of life and these gatherings may alleviate pandemic-related stress and anxiety [17] and enhance resilience [18], they also pose a risk of infection for participants.

The aforementioned characteristics give rise to an inquiry into the level of resilience demonstrated by various religious groups during the time when a substantial portion of the Israeli population had been vaccinated, resumed their regular routines, and perceived the arrival of the “day after” the pandemic with a reduced threat of infection. Our study aimed to investigate the impact of religiosity on resilience, anxiety, social support, and satisfaction with life during the initial stages of the conclusion of the third wave and the mass vaccination drive, one year into the pandemic.

We examined three hypotheses in the current research.

**Hypothesis** **1:**
*Firstly, it was hypothesized that the more religious the individual is, the higher their resilience, perceived social support, and satisfaction with life would be and the lower their anxiety.*


**Hypothesis** **2:**
*Secondly, we hypothesized that higher satisfaction with life and social support and a higher degree of religiosity would predict higher resilience.*


**Hypothesis** **3:**
*Finally, it was hypothesized that higher satisfaction with life, social support, a higher degree of religiosity, and higher resilience would predict lower anxiety.*


## 2. Theoretical Background

### 2.1. Resilience

Resilience pertains to the ability of individuals who are exposed to a potentially highly disruptive event, such as the death of a close relative or a life-threatening situation, to maintain mental health and psychological and physical functioning [19] and to reintegrate after challenging life experiences [20].

The stress-inducing characteristics of COVID-19 include uncertainty, ambiguity, loss of control, social isolation, and worries about one’s own health and that of loved ones [21]. Each of these is known to trigger stress and emotional distress, including internalizing symptoms such as anxiety and depression. In the face of COVID-19, one needs to cope with ongoing stressors and keep psychological distress to a minimum [22]. Resilience is related to well-being and is even in some cases measured by well-being instruments [23]. During times of need of resilience, religiosity plays an important role in an individual’s coping and ability to recover from stressful events by providing support in the face of adversity [24]. Comparative research conducted during COVID-19 among the population of Israel, Germany, Poland, and Slovenia revealed that Israeli youth exhibited lower levels of mental health difficulties compared to their counterparts in the other countries. This led the researchers to suggest that Israeli youth may possess greater mental resilience [25].

### 2.2. Well-Being (Satisfaction with Life)

Satisfaction with life refers to a person’s subjective evaluation of their overall well-being and happiness, as well as the quality of different aspects of their life. Various factors contribute to this assessment, including employment, leisure activities, housing and living conditions, family and social connections, as well as personal beliefs and religious practices [26].

Well-being is defined as a positive emotional state resulting from the harmony between specific contextual factors on the one hand and personal needs and expectations on the other hand [27] and a subjective feeling of health and positive perception of one’s quality of life [28]. Facing a worldwide pandemic may alter one’s sense of well-being and affect the ability to cope with the stressful situation imposed by the pandemic.

Research shows that positive emotions and a sense of well-being facilitate resilience because they both encourage adaptive coping [29] and the maintenance of social relationships [19]. Well-being was found to be negatively correlated with a sense of danger and distress symptoms [18] and positively related to higher levels of resilience [29]. The examination of well-being and its correlates in Israel during COVID-19 showed that symptoms of distress are related with lower emotional well-being and both well-being and resilience predict distress at that time [2].

### 2.3. Social Support

When individuals confront daily problems or major life stresses, they turn to others in their social network who can provide information, comfort, perspective, and help. These resources reinforce the ability to cope effectively and to reduce the destructive effects of stress [19], and they are associated with greater resilience [29,30]. Moreover, both social isolation and loneliness are associated with an increased risk of morbidity and mortality [31]. One of the direct outcomes of the pandemic was coercive social isolation that hindered people’s ability to seek social support, especially at times most needed, and circuitously may harm their resilience.

### 2.4. Anxiety during Crisis Situations

One of the most challenging emotions that tended to arise during COVID-19 was anxiety. The COVID-19 pandemic was a threat to all humans, undermining people’s basic sense of safety. People were likely to experience uncertainty and worry that they or their loved ones might become sick or even die [2]; they therefore might exhibit symptoms of anxiety and depression [21]. Furthermore, in a situation such as this, many individuals may wonder whether they were already affected by this virus although they did not show the typical symptoms caused by it [32]. The COVID-19 situation has generated feelings of uncertainty, existential threat, and ambiguous danger, which can trigger ongoing anxiety. A study conducted among the adult population in Israel aimed to investigate the risk factors associated with depression and anxiety during the pandemic. The study revealed a noteworthy increase in the prevalence of anxiety and depression symptoms among respondents. The author identified certain demographic risk factors, including younger age, female gender, lower income, lower education, single status, and the presence of a chronic medical condition. Moreover, fear of infection, social isolation, and economic adversity were found to significantly increase the risk of experiencing anxiety and depression [33]. In a situation of high stress and anxiety, higher resilience was found to decrease anxiety [2].

### 2.5. Religiosity and Faith

Religiosity may be a powerful resource for mental health, as it provides the individual with a framework of meaning-making associated with decreased psychological distress and a source of psychological well-being based on values [11]. It also provides the individual with attitudes and cognitions that can reframe negative events into less stressful events [34]. Religiosity contains a framework for behaviors, such as prayer, study, and mindfulness, that are related to lowered negative affect [35], and the religious community is a source of social support [11]. Those who are more religious have been found to have higher resilience against depression and anxiety disorders [34]. Research shows that during a time of higher mortality in the general population, such as a pandemic, religious beliefs intensify [36].

Religions instantiate many of the same protective systems implicated in resilience research. For example, humans appear to form attachment-like relationships with spiritual figures and religious leaders that may provide a secure base analogous to a parent attachment. Religious beliefs and practices also mobilize many of the adaptive systems, such as self-regulation through prayer or meditation, or social support and regulation through rituals, ceremonies, and rules for living [37]. The need to feel safe and secure is a basic human need provided at infancy by an attachment to a parent and fulfilled by other sources or entities when growing up [38,39].

In a time such as the COVID-19 pandemic when people are required to face an uncertain future with little control over what happens next, the only certainty for many is their religious faith [40], which is a rock against lean. Indeed, religious faith will likely make an important difference in how people will make it through this challenging time.

### 2.6. The Religious Groups in Israel

The Jewish population of Israel includes four subgroups that make up Israeli Jewry which are self-defined by the individual. Stolz and Usunier’s [41] definition of religious groups, which considers the degree of religiosity at the individual degree, is used to define the religious sub-groups in Israel based on the degree of active involvement of religion in their overall group culture. Nearly all Israeli Jews identify with one of these four categories on a spectrum between religious and secular: ultra-Orthodox as the most religious degree on the spectrum, religious, observant, and secular, which is the least degree of religiosity on the spectrum [42]. Highly religious (ultra-Orthodox) and secular Jews inhabit largely separate social worlds, with relatively few close friends and little intermarriage outside their own groups. The vast majority of observant and secular Jews say that democratic principles should take precedence over religious law, while a similarly large share of ultra-Orthodox say religious law (*halakha*) should take priority [43].

Religious Jews are usually modern individuals, well integrated into Jewish Israeli society. They participate in major institutions such as military, or secular post-secondary studies. They have high labor force participation rates, both men and women [44]. They form communities of their own that replicate in religious versions some structures instituted by the secular Jews [45].

Observant Jews do not define themselves as religious or ultra-Orthodox and not as secular. Usually, they do fulfill some religious commandments and traditions considered to be a sign of traditional beliefs. Their traditional behavior may be associated with identification and affiliation with the Jewish people, or with their Jewish ethnic community [44].

Israel’s ultra-Orthodox population is characterized by a relatively large number of children per family and high poverty rates [46] and a communal familial and social culture that values interaction with others [47]. Social support in these communities is an important mediator between religion and mental and physical health [34], despite the fact that it does not seem that this population has any greater access to medical services or organizations, compared to other groups in Israel [48]. Therefore, the orders to isolate and stay at home may be particularly stressful for these populations [17].

The ultra-Orthodox in Israel were acknowledged as a vulnerable group at risk for contracting COVID-19 [12,49]. Although they comprise 10% of the population, they accounted for more than one third of confirmed cases of COVID-19 and 60–70% of Israel’s COVID-19 hospitalizations during 2020. Furthermore, ultra-Orthodox towns constitute 6% of all the towns in Israel, while the rate of COVID-19 cases in these towns was 20.5% and kept rising. Thus, although on one hand religiosity seems to be helpful in alleviating levels of anxiety and stress, on the other hand it can have negative consequences in this population with regard to COVID-19 [14]. Due to the large families, overcrowded accommodations, and reduced social distance, the ultra-Orthodox may be exposed to the virus more than others [50]. During the first and the second lockdowns the ultra-Orthodox community had the highest levels of infection and mortality than other communities [16].

### 2.7. The Current Study

Resilience, as the ability to recover following a period of exposure to traumatic experience, may play an important role not only in the peak of the crisis but also in regaining or attaining effective or normal functioning [51], especially after a year of continuing stress.

Given the central role of resilience in emotional recovery and the place of religious faith when facing emotional challenges, our goal was to investigate how resilience relates to other constructs such as well-being, social support, and anxiety. Therefore, religiousness was defined as an independent factor, and resilience, and the above constructs were defined as dependent factors.

As far as we know, studies that have examined resilience and emotional factors in cultural groups in Israel have referred mainly to the first or second lockdown [52] of the COVID-19 pandemic. Another study was conducted among the American Orthodox Jewish population with regard to their distress during COVID-19 [17].

None of these studies examined the role of resilience in returning to regular life and routines at the end of the vaccination process in Israel among all the groups on the continuum of degrees of religion (i.e., ultra-Orthodox, religious, observant, secular) and the effect of resilience on top of the impact of demographic characteristics, such as gender, age, and SES. Israel was a leading country in vaccinating the population and therefore provided an opportunity to examine the resilience of the population at that unique point in time during the pandemic. We also examined the contribution of social support and well-being to resilience and anxiety in Israel alongside demographic characteristics.

This study utilized correlational research as its methodological approach. Three hypotheses were examined:

**H1:** 
*We hypothesized that due to religious beliefs, the more religious the individual is, the higher their resilience, perceived social support, and SWL would be and the lower their anxiety.*


**H2:** 
*It was hypothesized that higher SWL and social support and a higher degree of religiosity would predict higher resilience.*


**H3:** 
*It was hypothesized that higher SWL, social support, a higher degree of religiosity, and higher resilience would predict lower anxiety.*


## 3. Materials and Methods

### 3.1. Participants and Sampling

The total sample included 993 native Hebrew-speaking adults in Israel, from the ages of 18 years to 70 years (M = 40 years, SD = 13.96); 48% of them were male and 52% female. No difference was found between religiosity groups in the ratio of male to female (χ^2^ = 0.49, *p* = ns) or in age (*F* = 1.51, ns). The sample consisted of 88 (9%) ultra-Orthodox participants, with 66% of them below average SES; 106 (11%) religious participants, with 55% of them below average SES; 333 (34%) observant participants, with 62% of them below average SES; and 465 (47%) secular participants with 44% of them below average SES, χ^2^ = 32.01, *p* < 0.001. The distribution of religiosity groups in the sample represented the distribution of religiosity of the Jewish population in Israel [13] The ultra-Orthodox and the religious groups are minorities in the Jewish population in Israel (10% and 12%, respectively), and observant and secular participants are the majority groups (33% and 45%, respectively). The sample was recruited from an Israeli online panel (“Panel4all”) that consisted of about 80,000 Israeli participants who agreed to be registered to the panel and receive an online survey from time to time. Data were collected during one week in the middle of March 2021.

### 3.2. Research Instruments

#### 3.2.1. Resilience (CD-RISC) 

The questionnaire consists of ten items [53] taken from a 25-item resilience scale and includes one factor of resilience (a sample item: “I am able to adapt when changes occur”). Averaging to one continuous score, all items are rated on a Likert scale ranging from 1 (indicating the lowest level of agreement) to 5 (indicating the highest level of agreement).The internal reliability in the current research was 0.92.

#### 3.2.2. Generalized Anxiety Disorder Screener (GAD-7) 

The Generalized Anxiety Disorder (GAD-7) questionnaire is a seven-item [54], self-report anxiety questionnaire designed to assess the respondent’s health status. The items enquire about the degree to which the respondent has felt “nervous, anxious or on edge” or similar emotions (a sample item: “Becoming easily annoyed or irritable”). The items are rated on a 1- (not at all) to 4-point (nearly every day) scale of agreement. All the items are averaged to one continuous score. all items are rated on a Likert scale ranging from 1 (indicating the lowest level of agreement) to 5 (indicating the highest level of agreement).The internal reliability in the current research was 0.95.

#### 3.2.3. Satisfaction with Life Scale (SWLS) 

The SWLS is a widely used five-item index [55] scaled on a 1 (Strongly disagree) to 7 (Strongly agree) point scale, designed to measure global cognitive judgments regarding SWL and well-being (a sample item: “I am satisfied with my life.”). Total satisfaction with life is the average continuous score of all the items. The internal reliability in the current research was 0.90.

#### 3.2.4. Multidimensional Scale of Perceived Social Support (MSPSS) 

The MSPSS is a widely used questionnaire designed to assess perceptions of social support adequacy from three specific sources: family, friends, and significant others [56]. The questionnaire includes 12 items (a sample item: “I have a special person who is a real source of comfort to me”) rated on a 5-point Likert scale of agreement, ranging from 1 (strongly disagree) to 5 (strongly agree). There are four items in each subscale of social support, and the total perceived social support is determined by calculating the average continuous score of all items. The internal reliability in the current research was 0.94.

### 3.3. Data Analysis

Differences between groups were analyzed using Analysis of Variance, followed by Tukey post hoc comparisons for significant differences. The dependent variables for this analysis were resilience, anxiety, satisfaction with life, and social support. The independent variable was the degree of religiosity.

To test the second hypothesis regarding the prediction of resilience, a hierarchical linear regression with two forced steps was conducted. The first step included demographic variables as predictors to control for possible correlations, while the second step included social support and satisfaction with life (SWL) as predictors. The aim was to examine the amount of explained variance that the addition of these predictors contributed to the model. Similarly, in the regression predicting anxiety (third hypothesis), the same predictors as above were included in the first step. In the second step, resilience was added as an additional predictor to assess its contribution to the explained variance.

Differences in demographic characteristics between the groups of varying religiosities were examined using the chi-squared test. Data were analyzed using IBM SPSS, version 27.

## 4. Findings

### 4.1. Differences between Groups of Varying Religiosity (H1)

The first hypothesis addressed the differences in resilience, SWL, social support, and anxiety according to degree of religiosity. It was expected that higher degrees of religiosity would be characterized by higher resilience, SWL, and social support, and lower anxiety.

The findings pointed to significant differences between the groups of religiosity in resilience, *F*(3,989) = 10.29, *p* < 0.001, anxiety, *F*(3,989) = 10.73, *p* < 0.001 and SWL, *F*(3,989) = 18.67, *p* < 0.001 (see Table 1). According to the Tukey post hoc comparisons, ultra-Orthodox respondents scored higher on resilience than each of the other three groups. Religious and observant groups did not differ from each other.

Regarding anxiety, the results of post hoc (Tukey) tests indicated a lower level of anxiety among the ultra-Orthodox compared with each one of the other groups. The post hoc comparison of SWL showed that secular participants scored lower than each of the other three groups, and observant participants were lower in their SWL than ultra-Orthodox participants. No differences were found between the groups in perceived social support.

### 4.2. Prediction of Resilience (H2)

According to the second hypothesis, we expected that high degrees of religiosity and the emotional protective factors of SWL and social support would predict higher levels of resilience. The socioeconomic status of participants, which was related to the degree of religiosity and dependent variables (see Table 2), was included in the regression in addition to participants’ gender. The regression analysis revealed that low SES was associated with a lower level of resilience. In the first step, all three degrees of religiosity were related to higher resilience scores with a total explained variance of 4%. In the second step, when adding SWL and social support to the model, the percentage of variance was 31%, showing that higher SWL and social support predict higher resilience. Out of the four degrees of religiosity, only the ultra-Orthodox group was found to consistently exhibit higher levels of resilience. This result may be attributed to the covarying relationships between satisfaction with life and other variables that were introduced in this study.

### 4.3. Prediction of Anxiety (H3)

The results of the regression (see Table 3) showed that in the first step, older participants reported lower levels of anxiety. Low SES participants had a higher level of anxiety, as did female respondents. In addition, ultra-Orthodox religiosity was related to lower levels of anxiety. These background characteristics accounted for about 8% of the explained variance of anxiety. In the second step, the addition of SWL, social support, and resilience to the model added about 13% of variance, showing that higher SWL and a higher level of resilience were related with less anxiety.

## 5. Discussion

The aim of the current study was to examine the resilience and well-being of religious groups in Israel during the mass vaccination process to COVID-19. COVID-19 period was characterized by higher levels of anxiety and depression than before the pandemic in the Israeli population [33]. At the start of 2021, the State of Israel began a pioneering mass vaccination of the population [57,58], which soon resulted in getting back to routine life. In parallel, after three complete lockdowns, the population of Israel faced a decrease in their economic situation. As the health threat decreased but the economic threat posed a challenge to the recovery process, we were presented with a unique opportunity to examine both the factors that might relate to resilience as well as the concomitant anxiety level.

At first, we compared the level of resilience, SWL, social support, and anxiety among four groups of varying religious degrees. Then, we aimed to examine how emotional factors and demographic characteristics along with religiosity degree predict resilience and anxiety among the participants during this unique time.

The first hypothesis postulated that greater religiosity would be associated with higher levels of resilience, perceived social support, and satisfaction with life, as well as lower levels of anxiety, due to religious beliefs (i.e., ultra-Orthodox, religious, observant, and secular participants). The expected findings were consistent with previous research [59], indicating that individuals with higher levels of religiosity would report greater resilience, SWL, social support, and less anxiety. Similarly to other studies conducted during COVID-19 [24], Ultra-Orthodox participants exhibited higher levels of resilience. Their well-being as reflected by their satisfaction with life was higher as well, and they reported lower levels of anxiety than other less religious respondents. These results are supported by research that compared religious and secular people in regular times [43], and it was found that religious individuals had higher resilience in the wake of depression and anxiety disorders. A study conducted during the second wave of the COVID-19 pandemic in Israel revealed that ultra-Orthodox participants experienced higher resilience levels than secular participants [18].

It might be that religiosity is a resource of resilience since it involves a framework of meaning-making, a faith-based worldview, and trust in divine providence, all of which are associated with decreased psychological distress and a value-based pursuit of psychological well-being [11]. It enhances gratitude and includes practices and behaviors such as prayer, study, and mindfulness, which are linked to lower negative affect [35] and reduced depression and anxiety [34]. It has been suggested that religious faith and trust in God lead to experiencing life as meaningful, which in turn leads to improved mental health [34]. The faith-based worldview as a source of a high level of hope and purpose in life may explain the higher SWL that was found in our study among ultra-Orthodox respondents. The authors of a recent study conducted among the ultra-Orthodox community in Israel [24] suggest that adherence to traditional beliefs and practices and active involvement in one’s religion can provide comfort and hope in times of need, leading to improvements in well-being. In addition to their faith-based worldview, the ultra-Orthodox are identified as a religious collectivist community with very high levels of social capital relative to other populations in Israeli society [18]. The traditional structure of the ultra-Orthodox family tends toward large families with many children living in dense communities [47]. Chernichovsky and Sharoni [48] found that the physical health of the ultra-Orthodox population is better than would be expected based on their SES. Among this population, social capital affects health mainly through psychosocial support, including community aid, which reduces mental stress.

Despite the governmental social distancing guidelines in response to the pandemic, the ultra-Orthodox population maintained, as much as possible, their social and communal customs such as group Talmudic study in Yeshiva and praying three times daily in synagogues in groups of at least 10 people (a “Minyan”). Because of the traditional intimacy of their religious practices, the ultra-Orthodox population had an increased risk of contracting COVID-19 [49]. Therefore, it is not surprising that the death rate among this population was twice as high as in the general Jewish population [12]. Despite the high mortality rate, the strong belief in God and the practicing of religious rituals, as usual, seemed to be helpful in alleviating feelings of stress and anxiety from the pandemic, thus enhancing resilience [14]. The differences found between the groups at this specific point of time when the threat of infection seemed to be receding emphasize the need on the part of secular, observant, and religious populations for support and readjustment to routine life after COVID-19, as their resilience compared to the ultra-Orthodox is low.

The second hypothesis postulated that high degrees of religiosity and the emotional protective factors of SWL and social support will predict a higher level of resilience. Not surprisingly, higher SWL as well as greater social support were related to higher resilience. A study conducted during the first wave of COVID-19 in the US supports our findings and shows that individual resilience was greater among those who tended to get outside more often, exercise more, perceive more social support from family, friends, and significant others, sleep better, and pray more often [30]. Social support may nurture relationships with others and may offset the mental health challenges imposed by the COVID-19 pandemic. In addition, the ultra-Orthodox group exhibited higher levels of resilience on top of demographic characteristics, a finding that indicates that spiritual health may be another facet of SWL [30]. Our findings extend these conclusions to a time of coming back to regular life. In addition, although all levels of faith seem to explain resilience (see the first step of regression), the addition of social support to the statistical model (see the second step of regression) eliminated the effect of other religiosity levels (i.e., the religious and observant groups). Therefore, it is possible that the effect of religiosity is due to a combination of social support along with deep faith [37]. This possibility requires further study.

The ultra-Orthodox population in Israel is characterized by a low SES [48]. Therefore, one might have expected them to show higher levels of anxiety, with poor resilience in dealing with crises such as the pandemic [50]. Our findings indicate the opposite. The high level of resilience and low level of anxiety shed light on the central role of religion in impacting health outcomes among closed religious communities [46].

The third hypothesis aimed to examine the contribution of SWL and social support, degree of religiosity, background characteristics (SES, gender, and age) and the contribution of resilience to anxiety. We hypothesized that high degrees of religiosity and the emotional protective factors of SWL and social support, along with high resilience, would predict lower levels of anxiety. The literature indicates that participation in social activities is associated with better mental health [19]. The social isolation and distancing required to avoid the danger of infection may also contribute to higher levels of anxiety [17].

Therefore, the lack of contribution of social support to anxiety in the current study was a surprise. A possible explanation may be that in the case of the COVID-19 pandemic, anxiety may be evoked more by existential fears or worries about one’s health or the health of others [2], and therefore social support did not have a central role in reducing anxiety. It may be argued that in the unique situation of the pandemic, frequent interactions with friends and family may play a role in triggering anxiety.

In the current study, we examined how resilience was related to degree of religiosity, social support, and SWL. Research shows that religiosity and spirituality can function as a source of strength in facing challenges and difficult situations [60].

In this study, we explored how emotional and resilience factors contribute to the process of recovery and getting back to a routine when the health threat which elicited the crisis declined. The findings of this study provide quantitative evidence to the understanding of how religiosity level and emotional–social resources such as social support and satisfaction with life are related to resilience during a crisis and may have further implications on mental health.

In order to fully understand the role of resilience when dealing with crises and recovering from them, future research should examine additional factors that can explain resilience, such as further cultural differences that can point to populations that are at higher risk, such as other minority groups in Israel.

### 5.1. Implications for Practice

The global COVID-19 pandemic had profound impacts on individuals’ physical and emotional well-being, as well as their economic circumstances. Many people were confronted with distressing events and crises that were beyond their control. This study highlights that individuals’ coping mechanisms can vary according to their characteristics and that their level of resilience is a critical factor in determining their ability to cope effectively. The current study illustrates that religious faith, social support, and SWL play an important role in resilience and the ability to recover from a crisis. It points to the need for intervention programs that can foster and maintain emotional resilience by enhancing any kind of spiritual meaning and sources of social support.

### 5.2. Limitations

Our study was conducted at a unique period during the pandemic. However, baseline data for resilience and anxiety levels in the Israeli population were not present. For this reason, we were not able to ascribe the outcomes with certainty solely to the impact of the examined stressful situation of COVID-19. Furthermore, causal interpretation cannot be obtained based on the present research design; therefore, further studies could examine these issues with longitudinal data.

Another limitation concerns the sampling of ultra-Orthodox participants. The ultra-Orthodox in our study may be a representative sample of the modern ultra-Orthodox Jewish population in Israel (those who have computers or smartphones, study at universities, live outside of the core of the communities) and not the entire ultra-Orthodox population. Nevertheless, our findings highlight that for some communality and faith may serve as key factors in promoting resilience, especially during times of crisis. In the present study, we relied exclusively on self-report questionnaires to assess emotional variables, notably anxiety. However, it is acknowledged that incorporating physiological measures alongside self-reported measures can provide a more comprehensive understanding of emotional states. Regrettably, in the current study, we were unable to employ physiological measurements due to the perceived risk of viral transmission during the ongoing pandemic. As a result, it is recommended that future investigations explore the feasibility and advantages of integrating physiological measures in assessing emotional variables in comparable settings. This would enable researchers to elucidate the intricacies of emotional experiences and potentially enhance the validity of findings.

## Figures and Tables

**Table 1 behavsci-13-00398-t001:** M, SD, and F Values of Resilience and Emotional State by Degree of Religiosity.

		Degree of Religiosity					
		A Ultra-Orthodox	B Religious	C Observant	D Secular	F	Tukey Post Hoc Comparisons
Resilience	M	4.01	3.75	3.77	3.62	10.29 ***	A > B, C, D; C > D
	SD	0.68	0.56	0.64	0.66		
Anxiety	M	3.22	4.45	5.90	5.65	6.66 ***	A > B, C, D
	SD	4.56	5.10	5.70	5.85		
SWL	M	5.80	5.53	5.28	4.93	18.67 ***	D < A, B, C C < A
	SD	1.04	0.99	1.17	1.26		
Significant other support	M	6.3	6.08	6.05	6.08	1.07	
	SD	1.21	1.14	1.23	1.23		
Friend support	M	5.43	5.34	5.24	5.17	0.88	
	SD	1.52	1.5	1.52	1.65		
Family support	M	6.05	5.88	5.86	5.71	1.90	
	SD	1.34	1.23	1.38	1.46		

*** *p* < 0.001.

**Table 2 behavsci-13-00398-t002:** Pearson Correlation Coefficients Between Study Variables.

	1	2	3	4	5	6	7	8	9	10
1 Low SES	--									
2 High SES	−1.00 **	--								
3 Gender (female)	0.07 **	−0.08 **	--							
4 Ultra-Orthodox	0.08 **	−0.08 **	0.00	--						
5 Religious	0.01	−0.01	−0.02	−0.11 **	--					
6 Observant	0.12 ***	−0.12 ***	0.01	−0.22 ***	−0.25 ***	--				
7 Age	−0.05	0.05	0.00	−0.07 *	0.01	0.01	--			
8 Social support	−0.08 **	0.08 **	0.08 **	0.06 *	0.02	0.00	−0.02	--		
9 SWL	−0.13 ***	0.13 ***	0.02	0.16 ***	0.10 **	0.05 *	0.04	0.51 ***	--	
10 Resilience	−0.06 *	0.06 *	−0.05	0.14 ***	0.02	0.05 *	0.02	0.36 ***	0.54 ***	--
11 Anxiety	0.10 ***	−0.09 ***	0.12 ***	−0.12 ***	−0.06 *	0.06 *	−0.19 ***	−0.18 ***	−0.37 ***	−0.32 ***

* *p* < 0.05, ** *p* < 0.01, *** *p* < 0.001.

**Table 3 behavsci-13-00398-t003:** Hierarchical Regression Predicting Resilience and Anxiety by Demographic Variables (Standardized Beta Values).

Dependent Variable		Resilience	Anxiety
First step	Low SES	−0.08 **	0.06 *
	High SES	0.01	−0.07
	Gender (female)	−0.05	0.11 ***
	Ultra-Orthodox	0.18 ***	−0.15 ***
	Religious	0.06 *	−0.07 *
	Observant	0.12 ***	0.00
	Age	0.02	−0.19 ***
	R^2^	0.04 ***	0.08 ***
Second step	Low SES	0.01	0.01
	High SES	−0.01	−0.04
	Gender (female)	−0.07 **	0.11 ***
	Ultra-Orthodox	0.07 *	−0.06
	Religious	−0.02	−0.02
	Observant	0.04	0.06 *
	Age	0.01	−0.17 ***
	Social support	0.13 ***	0.02
	SWL	0.46 ***	−0.27 ***
	Resilience		−0.16 ***
	R^2^ total	0.31 ***	0.21 ***

* *p* < 0.01, ** *p* < 0.01, *** *p* < 0.001.

## Data Availability

The data presented in this study are not publicly available due to privacy restrictions.

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
