# Peer review of "Uncovering Resilience Disparities among Religious Groups in Israel’s Mass COVID-19 Vaccination Drive: Lessons Learned to the Post-COVID Era"

_behavsci, 2023, doi:10.3390/bs13050398_

Round 1
Reviewer 1 Report
please find attached

no more comment
Author Response
Thank the reviewer for these insightful comments. Please see attachment of our response.

Reviewer 2 Report
The paper raises important approaches to examine the emotional resilience, satisfaction with 9 life, social support, and anxiety during the vaccination process of the Israeli population after the end of the third lockdown, according to religiosity level. I have tried to assess the paper based on methodological and empirical contributions. Certain aspects could be improved by being more specific when describing and referencing the situation encountered, as well as the data that has already been evaluated by other research. Despite this, my congratulations to the authors, as it was certainly not an easy task to research.
-More information on the covid status and pre- and post-pandemic measurements is needed, in my opinion, and this should be included in the introduction. In addition, the section on mental health might be expanded with more references and maybe some comparative statistics from Israel and elsewhere. The article's hypotheses and the ultimate goal should also be included at the end, just as they do in the abstract.
-The description of the methods should be more concise or provide examples of the questions.
- I consider that the discussion should be validated and compared with many more references as there is a lot of content dedicated to covid and mental health.
- Aspects such as anxiety or stress could also have been measured with physiological variables which, not having been able to be applied, could be indicated as a limitation for the study. In addition, the data may have been biased by answering the following questions.
-Overall the manuscript can be better structured and improved with new citations. The wording is sometimes difficult to understand, perhaps it could be improved as well.
Author Response
We are grateful the reviewer for the insightful comments. Please see attached document with response.

Round 2
Reviewer 2 Report
The authors have greatly improved the manuscript, however I still advise that the hypotheses are still listed in the introduction and do not appear until 3.3.
I also believe that there are typographical errors in the references, especially those where web links appear. I think it is a general problem of not adapting to the style of lettering and references requested by the journal.
Otherwise, if this is modified, it is a correct manuscript.
Author Response
Thank the reviewer for these comment. Uploaded is the updated version of our manuscript, the changes are shadowed by blue color
